# Semi-Supervised Multimodal Learning with Deep Generative Models

**Masahiro Suzuki, Yutaka Matsuo**
The University of Tokyo
Bunkyo-ku, Tokyo, Japan
{`masa`,`matsuo`}@weblab.t.u-tokyo.ac.jp

## Abstract

In recent years, deep neural networks are used mainly as discriminators of multimodal learning. We should have large amounts of labeled data for training them, but obtaining such data is difficult because it requires much labor to label inputs. Therefore, semi-supervised learning, which improves the discriminator performance using unlabeled data, is important. Among semi-supervised learning, methods based on deep generative models such as variational autoencoders (VAEs) are known to be trained end-to-end with high accuracy. In this paper, we propose a novel model of semi-supervised multimodal learning based on multimodal VAEs: SS-HMVAE. Furthermore, to cope with unimodal inputs in test data, we propose an extended model based on existing studies of complementation of missing values, which we call SS-HMVAE-kl. From experimentation, we confirm that the proposed model has higher performance than either conventional unimodal or multimodal semi-supervised learning.

## 1 Introduction

We constantly interact with various kinds of information. Each is called a modality, and we are conducting more reliable information processing based on *multimodal* information. For machine learning in recent years, multimodal learning that treats multimodal information as inputs has been studied widely (Lahat et al., 2015; Baltrušaitis et al., 2017). The most common setting of multimodal learning is to predict labels from multimodal data as inputs, which is called fusion setting.

Recently, deep neural networks are often used as discriminators for fusion setting because of their high performance and ease of design (Ngiam et al., 2011). By sharing the top hidden layers of the networks for each modality and by training them, one can obtain a joint representation that integrates information of multiple modalities and that can be useful for predicting labels. In general, training of deep neural networks requires large labeled datasets. However, while the input data of each modality network can be obtained easily, it is difficult to obtain corresponding label information because human resources are required.

One approach to solving this difficulty is semi-supervised learning, which is a framework that improves the discriminator performance using not only labeled data but also large amounts of unlabeled data for training. Cheng et al. (2016) proposes semi-supervised multimodal learning by co-training using deep neural networks. In their framework, we can train not only a discriminator given all modalities as inputs but also discriminators given each modality as input. However, this method cannot be trained end-to-end. Moreover, it is necessary to devise various additional measures specialized for the dataset used for training.

Deep generative models can handle unlabeled and labeled data in a unified manner, and can execute semi-supervised learning end-to-end. Among them, methods based on variational autoencoders (VAEs) (Kingma & Welling, 2013) are known to have higher performance than that provided by conventional semi-supervised learning (Kingma et al., 2014; Maaløe et al., 2016).

Therefore, we propose a novel model of semi-supervised multimodal learning using deep generative models, which we call *Semi-Supervised Hierarchical Multimodal Variational AutoEncoder (SS-HMVAE)*. However, if inputting unimodal data at testing as did Cheng et al. (2016), then other

multimodal inputs should be missing, which can degrade accuracy. Therefore, we propose the additional approach by extending SS-HMVAE based on existing studies of complementation of missing values (Suzuki et al., 2016; 2018), which we call SS-HMVAE-kl.

## 2 PROBLEM FORMULATION

We assume a dataset $\mathcal{D}_{\mathcal{L}} = \{(\boldsymbol{x}_1, \boldsymbol{w}_1, \boldsymbol{y}_1), ..., (\boldsymbol{x}_N, \boldsymbol{w}_N, \boldsymbol{y}_N)\}$ given as a training set, where $\boldsymbol{x}$ and $\boldsymbol{w}$ are different modalities[1], and where $\boldsymbol{y} \in \{0, 1\}^K$ is label information representing those target categories. Each example of the dataset $(\boldsymbol{x}_n, \boldsymbol{w}_n, \boldsymbol{y}_n)$ represents the same object.

The challenge of semi-supervised multimodal learning in this study is to estimate discriminators not only in multimodal inputs, $p(\boldsymbol{y}|\boldsymbol{x}, \boldsymbol{w})$, but also in unimodal inputs, $p(\boldsymbol{y}|\boldsymbol{x})$ and $p(\boldsymbol{y}|\boldsymbol{w})$, from a small number of labeled set $\mathcal{D}_{\mathcal{L}}$ and a large number of unlabeled set $\mathcal{D}_{\mathcal{U}} = \{(\boldsymbol{x}_1, \boldsymbol{w}_1), ..., (\boldsymbol{x}_M, \boldsymbol{w}_M)\}$.

## 3 PROPOSED METHOD

Let $\boldsymbol{y} \sim p(\boldsymbol{y}) = \mathrm{Cat}(\boldsymbol{y}; \boldsymbol{\pi})$, $\boldsymbol{z} \sim p(\boldsymbol{z}) = \mathcal{N}(\boldsymbol{0}, \boldsymbol{I})$, $\boldsymbol{a} \sim p_{\boldsymbol{\theta}}(\boldsymbol{a}|\boldsymbol{z}, \boldsymbol{y})$, $\boldsymbol{x}, \boldsymbol{w} \sim p_{\boldsymbol{\theta}}(\boldsymbol{x}, \boldsymbol{w}|\boldsymbol{a})$ be generative processes of modalities $\boldsymbol{x}, \boldsymbol{w}$ and a label $\boldsymbol{y}$, where $\boldsymbol{z}$ and $\boldsymbol{a}$ are latent variables and $\boldsymbol{\theta}$ is a parameter of each generative model. At this time, the joint distribution of all modalities and a label becomes $p(\boldsymbol{x}, \boldsymbol{w}, \boldsymbol{y}) = \int \int p_{\boldsymbol{\theta}}(\boldsymbol{x}|\boldsymbol{a}) p_{\boldsymbol{\theta}}(\boldsymbol{w}|\boldsymbol{a}) p_{\boldsymbol{\theta}}(\boldsymbol{a}|\boldsymbol{z}, \boldsymbol{y}) p(\boldsymbol{z}) p(\boldsymbol{y}) d\boldsymbol{a} d\boldsymbol{z}$.

Training this deep generative model requires maximization of this joint distribution over a training set. However, perform this maximization directly is difficult because this distribution is intractable. Therefore, we instead maximize the following evidence lower bound (ELBO).

$$\mathcal{L}(\boldsymbol{x}, \boldsymbol{w}, \boldsymbol{y}) = E_{q_{\boldsymbol{\phi}}(\boldsymbol{a}, \boldsymbol{z}|\boldsymbol{x}, \boldsymbol{w}, \boldsymbol{y})}[\log \frac{p_{\boldsymbol{\theta}}(\boldsymbol{x}|\boldsymbol{a}) p_{\boldsymbol{\theta}}(\boldsymbol{w}|\boldsymbol{a}) p_{\boldsymbol{\theta}}(\boldsymbol{a}|\boldsymbol{z}, \boldsymbol{y}) p(\boldsymbol{z}) p(\boldsymbol{y})}{q_{\boldsymbol{\phi}}(\boldsymbol{a}, \boldsymbol{z}|\boldsymbol{x}, \boldsymbol{w}, \boldsymbol{y})}], \tag{1}$$

where $q_{\boldsymbol{\phi}}(\boldsymbol{a}, \boldsymbol{z}|\boldsymbol{x}, \boldsymbol{w}, \boldsymbol{y}) = q_{\boldsymbol{\phi}}(\boldsymbol{z}|\boldsymbol{a}) q_{\boldsymbol{\phi}}(\boldsymbol{a}|\boldsymbol{x}, \boldsymbol{w})$ is an approximate distribution of a posterior, or inference model, and $\boldsymbol{\phi}$ represents its parameter. To optimize this ELBO with respect to parameters, we can estimate gradients of ELBO using stochastic gradient variational Bayes (SGVB) (Kingma & Welling, 2013; Rezende et al., 2014).

Next, we derive ELBO over an unlabeled dataset. Using the discriminative model $q_{\boldsymbol{\phi}}(\boldsymbol{y}|\boldsymbol{x}, \boldsymbol{w}) = E_{q_{\boldsymbol{\phi}}(\boldsymbol{a}|\boldsymbol{x}, \boldsymbol{w})}[q_{\boldsymbol{\phi}}(\boldsymbol{y}|\boldsymbol{a})]$, ELBO of the joint distribution of all modalities $p(\boldsymbol{x}, \boldsymbol{w})$ becomes as follows:

$$\mathcal{U}(\boldsymbol{x}, \boldsymbol{w}) = E_{q_{\boldsymbol{\phi}}(\boldsymbol{a}, \boldsymbol{z}, \boldsymbol{y}|\boldsymbol{x}, \boldsymbol{w})}[\log \frac{p_{\boldsymbol{\theta}}(\boldsymbol{x}|\boldsymbol{a}) p_{\boldsymbol{\theta}}(\boldsymbol{w}|\boldsymbol{a}) p_{\boldsymbol{\theta}}(\boldsymbol{a}|\boldsymbol{z}, \boldsymbol{y}) p(\boldsymbol{z}) p(\boldsymbol{y})}{q_{\boldsymbol{\phi}}(\boldsymbol{a}, \boldsymbol{z}, \boldsymbol{y}|\boldsymbol{x}, \boldsymbol{w})}], \tag{2}$$

where $q_{\boldsymbol{\phi}}(\boldsymbol{a}, \boldsymbol{z}, \boldsymbol{y}|\boldsymbol{x}, \boldsymbol{w}) = q_{\boldsymbol{\phi}}(\boldsymbol{z}|\boldsymbol{a}, \boldsymbol{y}) q_{\boldsymbol{\phi}}(\boldsymbol{y}|\boldsymbol{a}) q_{\boldsymbol{\phi}}(\boldsymbol{a}|\boldsymbol{x}, \boldsymbol{w})$. Then we use Gumbel-softmax (Jang et al., 2016) to reparameterize a categorical distribution $q_{\boldsymbol{\phi}}(\boldsymbol{y}|\boldsymbol{a})$.

Therefore, the objective $\mathcal{J}_{HMVAE}$ over both labeled and unlabeled sets is as follows:

$$\mathcal{J}_{HMVAE} = \frac{1}{N} \sum_{(\boldsymbol{x}_n, \boldsymbol{w}_n, \boldsymbol{y}_n) \in \mathcal{D}_{\mathcal{L}}} \mathcal{L}_l(\boldsymbol{x}_n, \boldsymbol{w}_n, \boldsymbol{y}_n) + \frac{1}{M} \sum_{(\boldsymbol{x}_m, \boldsymbol{w}_m) \in \mathcal{D}_{\mathcal{U}}} \mathcal{U}(\boldsymbol{x}_m, \boldsymbol{w}_m), \tag{3}$$

where $\mathcal{L}_l(\boldsymbol{x}, \boldsymbol{w}, \boldsymbol{y}) = \mathcal{L}(\boldsymbol{x}, \boldsymbol{w}, \boldsymbol{y}) + \alpha \cdot \log q_{\boldsymbol{\phi}}(\boldsymbol{y}|\boldsymbol{x}, \boldsymbol{w})$. $\alpha$ is a parameter that adjusts the ratio between discriminative and generative models in training.

Even if we optimize Equation 3, only $q_{\boldsymbol{\phi}}(\boldsymbol{y}|\boldsymbol{x}, \boldsymbol{w})$ is trained as a discriminative model. Therefore, to predict the label from unimodal input, we must miss another modality input. However, this missing input might adversely affect label prediction. One method of avoid such effects is application of a missing value complement technique such as iterative sampling method[2] (Rezende et al., 2014). However, this method has been shown to be unable to cope appropriately when the missing modality dimensions are numerous (Suzuki et al., 2018). Therefore, we extend SS-HMVAE using the same approach as JMVAE-kl [3], proposed to address the missing problem in multimodal VAEs (Suzuki et al., 2016; 2018). We call this approach *SS-HMVAE-kl*.

We prepare a new inference model for each modality, $q_{\boldsymbol{\lambda}}(\boldsymbol{a}|\boldsymbol{x})$ and $q_{\boldsymbol{\lambda}}(\boldsymbol{a}|\boldsymbol{w})$, where $\boldsymbol{\lambda}$ is a parameter of each model. If we can properly train them, we can obtain discriminative models of unimodal input, such as $q_{\boldsymbol{\phi}, \boldsymbol{\lambda}}(\boldsymbol{y}|\boldsymbol{x}) = E_{q_{\boldsymbol{\lambda}}(\boldsymbol{a}|\boldsymbol{x})}[q_{\boldsymbol{\phi}}(\boldsymbol{y}|\boldsymbol{a})]$. Therefore, we add divergence between these and

---

[1] In this paper, we limit the number of modalities to two.

[2] See the appendix for details on how to perform the iterate sampling method with SS-HMVAE.

[3] Vedantam et al. (2017) and Higgins et al. (2017) refer to JMVAE-kl simply as JMVAE.

Table 1: Comparison with existing semi-supervised learning of unimodal and multimodal. These results are averages for 10 different train/test splits. †These results were reproduced from the original papers.

|  | Models | RGB | depth | RGB+depth |
|---|---|---|---|---|
| Unimodal | M2 (Kingma et al., 2014) | $85.6 \pm 1.6$ | $72.0 \pm 1.7$ | - |
|  | SDGM (Maaløe et al., 2016) | $85.6 \pm 1.9$ | $75.8 \pm 1.7$ | - |
| Multimodal | CT+SVM† (Cheng et al., 2015) | 78.7 | 75.4 | 83.7 |
|  | Co-training† (Cheng et al., 2016) | $85.5 \pm 2.0$ | $\mathbf{82.6} \pm 2.3$ | $89.2 \pm 1.3$ |
|  | SS-MVAE | $79.6 \pm 1.8$ | $34.4 \pm 7.0$ | $89.9 \pm 1.7$ |
| Proposed | SS-HMVAE | $85.4 \pm 2.2$ | $41.5 \pm 6.7$ | $\mathbf{90.6} \pm 1.6$ |
|  | SS-HMVAE (iterative sampling) | $86.4 \pm 2.1$ | $54.8 \pm 1.9$ | $\mathbf{90.6} \pm 1.6$ |
|  | SS-HMVAE-kl | $\mathbf{86.8} \pm 2.2$ | $81.1 \pm 2.4$ | $90.2 \pm 1.4$ |

the inference model of SS-HMVAE to Equation 3 to approximate them more closely in training.

$$\mathcal{J}_{kl} = \mathcal{J}_{HMVAE} - \frac{\beta}{M+N}\mathcal{J}_{div}, \tag{4}$$

where $\beta$ is a parameter that adjusts the influence of the second term, and

$$\mathcal{J}_{div} = \sum_{(\boldsymbol{x}_n, \boldsymbol{w}_n) \in \mathcal{D}_{\mathcal{L}} \bigcup \mathcal{D}_{\mathcal{U}}} [D_{KL}(q_{\boldsymbol{\phi}}(\boldsymbol{a}|\boldsymbol{x}_n, \boldsymbol{w}_n)||q_{\boldsymbol{\lambda}}(\boldsymbol{a}|\boldsymbol{x}_n)) + D_{KL}(q_{\boldsymbol{\phi}}(\boldsymbol{a}|\boldsymbol{x}_n, \boldsymbol{w}_n)||q_{\boldsymbol{\lambda}}(\boldsymbol{a}|\boldsymbol{w}_n))].$$

By optimizing this objective of SS-HMVAE-kl, we can train discriminative models of multimodal and unimodal inputs by end-to-end.

## 4 EXPERIMENT

In this experiment, we use the Washington RGB-D dataset (Lai et al., 2011), which consists of color (RGB) and depth images, regarded as different modalities. Each example represents one of 300 household items, and they are grouped into 51 categories. According to Lai et al. (2011), about 35,000 examples were set as a training set and 6,877 as a test set. In addition, 5% of the training set was selected randomly to be a labeled set, and the rest were set as an unlabeled set. See the appendix for preprocessing of the dataset and the network structures of each distribution. The number of iterative sampling was set to 100, and we set $\alpha = \beta = 1$. We used Tars[4] to implement the models.

To evaluate the performance of the proposed methods, we compare them with existing semi-supervised learning of unimodal (M2 (Kingma et al., 2014), SDGM (Maaløe et al., 2016)) and multimodal (CT+SVM (Cheng et al., 2015), co-training (Cheng et al., 2016), SS-MVAE). SS-MVAE is simply a multimodal extension of M2, which is almost identical to semiMVAE (Du et al., 2017) [5]. Note that we cannot apply complementary methods such as SS-HMVAE-kl to SS-MVAE.

Table 1 presents the classification accuracies of respective models. First, compared to the proposed methods, the unimodal input accuracy is not much improved by the iterative sampling method, but it is greatly improved by SS-HMVAE-kl. This is a better result than those obtained using semi-supervised models of unimodal input. Next, compared with existing multimodal methods, the proposed models outperform them in multimodal input (RGB+depth). Even in the case of unimodal input (RGB, depth), the proposed methods almost outperform the existing ones. In depth, co-training is better than our methods, perhaps because of the difference in depth preprocessing. Furthermore, note that Cheng et al. (2016) uses not only co-training but also various techniques for performance enhancement.

## 5 CONCLUSION

In this paper, we focused on semi-supervised multimodal learning and proposed SS-HMVAE based on deep generative models. We also proposed SS-HMVAE-kl to cope with unimodal input. Results of experiments confirmed that the proposed models outperform existing models.

---

[4]https://github.com/masa-su/Tars. This is a deep generative model library in Theano (Team et al., 2016) and Lasagne (Dieleman et al., 2015).

[5]Actually, the only difference between semiMVAE and SS-MVAE is that semiMVAE sets a mixed Gaussian in the inference distribution, whereas SS-MVAE sets a single Gaussian.

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

## A  PREPROCESSING OF THE DATASET

First, we resize both RGB and depth images to $148 \times 148$ pixels. Because the original images are portrait or landscape, the longer side of the original images is fixed at 148, and the shorter side is interpolated by extending the edge pixel. Next, we interpolate the missing values of the distance in the depth images with the nearest distance values and normalize the distance values to $[0, 225]$. Furthermore, we extend the single channel depth images to three channels using the jet colormap process. All preprocessing above is done in accordance with Eitel et al. (2015), where only the image size follows Cheng et al. (2016). Note that the method described by Cheng et al. (2016) preprocesses depth images with surface normal processing, which provides higher accuracy than jet colormap.

In this experiment, we do not treat RGB-D image directly as an input of deep generative models. We use features extracted from deep neural networks as input because the purpose of this study is not to generate images. The deep neural network for feature extraction is pre-trained VGG16 (Simonyan & Zisserman, 2014) using the ILSVRC 2012 dataset. The output values at the fc1 layer (4096 dimensions) of it are used as input features. We prepared VGG16 for each modality, with fine-tuning only of the labeled set. We used Adam (Kingma & Ba, 2014) and trained 200 epochs with a learning rate of $10^{-5}$ to prevent over-fitting.

Therefore, the input features of RGB and depth images are $\boldsymbol{x} \in \mathcal{R}_{>0}^{4096}$ and $\boldsymbol{w} \in \mathcal{R}_{>0}^{4096}$ [6].

## B  PARAMETERIZATION OF DISTRIBUTIONS WITH DEEP NEURAL NETWORKS

The Gaussian distribution can be parameterized with deep neural networks as
$$\mathcal{N}(\mathbf{z}; \boldsymbol{\mu}, \mathrm{diag}(\boldsymbol{\sigma}^2)),$$
$$\boldsymbol{\mu} = f_\mu(f_{\mathrm{MLP}}(\mathbf{x})),$$
$$\boldsymbol{\sigma}^2 = \mathrm{Softplus}(f_{\sigma^2}(f_{\mathrm{MLP}}(\mathbf{x}))),$$
where $f_\mu$ and $f_{\sigma^2}$ are respectively denote linear single layer neural networks and where $f_{\mathrm{MLP}}$ represents a deep neural network with an arbitrary number of layers. Moreover, applying the softplus function for each element of a vector is denoted as $\mathrm{Softplus}$.

The Bernoulli distribution is parameterized as
$$p_\theta(\mathbf{x}|\mathbf{z}) = \mathcal{B}(\mathbf{x}; \boldsymbol{\mu}), \boldsymbol{\mu} = \mathrm{Sigmoid}(f_\mu(f_{\mathrm{MLP}}(\mathbf{z}))),$$
where $\mathrm{Sigmoid}$ is represents the sigmoid function.

In the case of the categorical distribution, we can parameterize it as
$$p_\theta(\mathbf{x}|\mathbf{z}) = \mathcal{C}(\mathbf{x}; \boldsymbol{\mu}), \boldsymbol{\mu} = \mathrm{Softmax}(f_\mu(f_{\mathrm{MLP}}(\mathbf{z}))),$$
where $\mathrm{Softmax}$ denotes the softmax function.

## C  MODEL ARCHITECTURE

For the notation of model structures, we denote a linear fully-connected layer with $k$ units, batch normalization, and ReLU as `DkBR`. Also, we denote `DkBR` without batch normalization and ReLU as

---

[6] These domains of definition become positive real numbers because the activation function of the fc1 layer is ReLU.

`Dk`. In addition, the process of applying `J` after `I` is denoted as `I-J`, and the process of concatenating the last layers of the two networks `I, J` into one layer is denoted as `(I,J)`.

Therefore, the network structures of distributions of SS-HMVAE are as follows:

- $p(\boldsymbol{x}|\boldsymbol{a}), p(\boldsymbol{w}|\boldsymbol{a})$
  - $f_\mu$: `D1024`
  - $f_{\mathrm{MLP}}$: `a-D1024BR-D1024BR`
- $q(\boldsymbol{y}|\boldsymbol{a})$ (categorical)
  - $f_\mu$: `D51`
  - $f_{\mathrm{MLP}}$: `a-D1024BR-Dropout0.5`
- $q(\boldsymbol{a}|\boldsymbol{x},\boldsymbol{w})$ (Gaussian)
  - $f_\mu$ and $f_{\sigma^2}$: `D1024`
  - $f_{\mathrm{MLP}}$: `(x-D1024BR, w-D1024BR)`
- $q(\boldsymbol{a}|\boldsymbol{z},\boldsymbol{y})$ (Gaussian)
  - $f_\mu$ and $f_{\sigma^2}$: `D1024`
  - $f_{\mathrm{MLP}}$: `(z-D1024BR, y-D1024BR)`
- $q(\boldsymbol{z}|\boldsymbol{a},\boldsymbol{y})$ (Gaussian)
  - $f_\mu$ and $f_{\sigma^2}$: `D1024`
  - $f_{\mathrm{MLP}}$: `(a-D1024BR, y-D1024BR)`

where `DropoutRate` denote the dropout layer with the dropout rate `Rate`.

Furthermore, the inference models of each modality of SS-HMVAE-kl are set as follows:

- $q(\boldsymbol{a}|\boldsymbol{x}), q(\boldsymbol{a}|\boldsymbol{w})$ (Gaussian)
  - $f_\mu$ and $f_{\sigma^2}$: `D1024`
  - $f_{\mathrm{MLP}}$: `x or w-D1024BR`

We used Adam for the optimization algorithm. The batch size was 128, the learning rate was $10^{-4}$. Then we trained 200 epochs.

## D    ITERATIVE SAMPLING IN SS-HMVAE

SS-HMVAE contains the latent variable $\boldsymbol{a}$. This variable plays the role of a joint representation integrating multimodal information. Therefore, when the input $\boldsymbol{x}$ of the discriminative model is missing, the transition kernel can be written using $\boldsymbol{a}$ as follows:

$$T(\tilde{\boldsymbol{x}}|\boldsymbol{x},\boldsymbol{w}) = \int p(\tilde{\boldsymbol{x}}|\boldsymbol{a})q(\boldsymbol{a}|\boldsymbol{x},\boldsymbol{w})d\boldsymbol{a} \tag{5}$$

Therefore, the processes of the iterative sampling method are described below: First, let the initial value of $\boldsymbol{x}$ be random noise such as $\boldsymbol{x} \sim p(\boldsymbol{x})$. We then sample $\boldsymbol{a}$ using the inference model $q(\boldsymbol{a}|\boldsymbol{x},\boldsymbol{w})$ and reconstruct $\boldsymbol{x}$ by sampling from the generative model $p(\boldsymbol{x}|\boldsymbol{a})$.

By repeating these processes several times, the missing modality $\boldsymbol{x}$ becomes supplemented.

