# OpenReview forum: "Semi-Supervised Multimodal Learning with Deep Generative Models"
_ICLR.cc/2018/Workshop — Reject_

### Official Review · AnonReviewer2 · 2018-02-26
**Very dense, bit difficult to read, experimental results seem ok**

**Rating:** 6
**Confidence:** 2

**Review:**

The idea is to have a VAE for multimodal data . for semi-supervised learning.
A generative model is defined, starting by defining a distribution of the labels to condition the generative process of the two modalities.

The description of SS-HMVAE-kl is so concise that it becomes unclear to me.

There are a couple of sentences (second to last paragraph on P2) that need updating.
The experimental results appear to be good, but I have difficulties understanding what is actually been done.

I believe the current version to be good enough for workshop acceptance.
I would recommend the authors to update the wording in the manuscript.

---

### Official Review · AnonReviewer3 · 2018-03-10

**Rating:** 4
**Confidence:** 4

**Review:**

This paper proposed SS-HMVAE and SS-HMVAE-kl for semi-supervised multimodal learning. Specifically, the models proposed in this paper are quite similar to M2 of Diederik P Kingma 2014 with the difference in that multimodal data are considered. Hence I think the novelty is incremental. Moreover, it seems the decomposition of q_{\phi}(a,z|x,w,y) under equation 1 is incorrect, since y is missed. As for the experimental results, the improvement over other baselines is not significant.

---

### Official Review · AnonReviewer1 · 2018-03-12
**not exciting enough for an iclr workshop paper**

**Rating:** 5
**Confidence:** 3

**Review:**

Summary:
This paper proposes a VAE model for semi-supervised multimodal learning--that is semi-supervised learning when the covariates have different domains of definition. The model is extended to deal with the problem of missing covariate at test time.

Quality:
The maths is correct.

Clarity:
The paper can benefit from a lot of editing but the overall goal and idea are clear.

Originality and Significance:
It is marginal. See Suzuki et al 2016

---

### Decision · Program_Chairs · 2018-03-20
**ICLR 2018 Workshop Acceptance Decision**

**Decision:**

Reject

**Comment:**

Based on the reviews, this paper has not been accepted for presentation at the ICLR workshop. However, the conversation and updates can continue to appear here on OpenReview.